# ChatPathway: Conversational Large Language Models for Biology Pathway Detection

## Abstract

Biological pathways, like protein-protein interactions and metabolic networks, are vital for understanding diseases and drug development. Some databases such as KEGG are designed to store and map these pathways. However, many bioinformatics methods face limitations due to database constraints, and certain deep learning models struggle with the complexities of biochemical reactions involving large molecules and diverse enzymes. Importantly, the thorough exploration of biological pathways demands a deep understanding of scientific literature and past research. Despite this, recent advancements in Large Language Models (LLMs), especially ChatGPT, show promise. We first restructured data from KEGG and augmented it with molecule structural and functional information sourced from UniProt and PubChem. Our study evaluated LLMs, particularly GPT-3.5-turbo and Galactica, in predicting biochemical reactions and pathways using our constructed data. We also assessed its ability to predict novel pathways, not covered in its training dataset, using findings from recently published studies. While GPT demonstrated strengths in pathway mapping, Galactica encountered challenges. This research emphasizes the potential of merging LLMs with biology, suggesting a harmonious blend of human expertise and AI in decoding biological systems.

## 1 Introduction

Biological pathways, including protein–protein interaction networks, metabolic networks, and gene regulatory networks, are intricate systems of proteins and molecules interacting in processes like signaling [24]. Investigating these pathways is essential for understanding disease mechanisms and drug development. Techniques like GSEA [25] and SPIA [26] aid this exploration. Databases like KEGG [13] have also been developed to collate and visualize these networks systematically. However, identifying pathways from genome-inferred proteins and enzymes is limited by database completeness, complicating data integration from varied sources.

Recent advancement in deep learning offers new tools for scientific research. Among these tools, Large language models (LLMs) have been revolutionizing in the domain of Natural Language Processing (NLP). Their fantastic generalization ability is obtained by pretraining on extensive textual datasets. ChatGPT, released by OpenAI in November 2022 [22], exemplifies the capabilities of such models. ChatGPT's aptitude for understanding scientific texts has seen its application in interpreting protein sequences and molecular structures like SMILES, aiding in tasks such as molecule editing [18] and chemistry-focused projects [6]. Other models, such as SciBERT [5], BioLM [17], and Galactica [27], have been developed explicitly for processing scientific literature and typically of a smaller scale.

Submitted to NeurIPS 2021 AI for Science Workshop.

Existing deep learning methods mainly utilize chemical structure in the format of graph [8, 28] or SMILES [20] to predict chemical reactions. Predicting biochemical reactions poses a more complex set of challenges compared to chemical reactions. A significant factor is the involvement of large molecules, which often participate in these reactions. These large molecules introduce an additional layer of complexity due to their diverse structures and functionalities, necessitating a more nuanced approach for accurate prediction. Besides, the outcome of biochemical reactions is also significantly influenced by the variety of enzymes involved. Different enzymes have the potential to catalyze the same set of reactants to yield diverse products, following unique pathways and mechanisms. This variability and diversity necessitate a comprehensive understanding of enzyme specificity and the conditions under which they operate. Yet, LLMs can integrate data from diverse sources, offering nuanced insights into pathway interactions. Given their training in vast data repositories and their ability of reasoning, they may deduce interactions based on existing knowledge or even pioneer groundbreaking hypotheses about pathways, which helps experimental design in wet-lab.

Consequently, our research seeks to assess the capabilities of LLMs for pathway predictions. We have formulated three primary tasks for our assessment in Figure 1: (1) *Biochemical Reaction Prediction*: Given reactants and enzymes, the task is to predict the products. (2) *Metabolism Pathway Prediction*: This involves predicting the complete metabolic pathway given the initial reactants and enzymes, where each step is similar to the biochemical reaction prediction. (3) *Regulatory Pathway Prediction*: The objective is to predict the complete regulatory relationships within a pathway given stimuli. These tasks are framed within the context of identifying the relationships between the input elements and predicting the likely outputs, providing a basis for understanding the underlying probabilistic distributions of biochemical reactions and pathways. To accomplish these tasks, we systematically restructured relevant data from KEGG through the API, incorporating important functional and structural information retrieved from UniProt [1] and PubChem [16]. We utilized data encompassing 11,944 reactions, 480 metabolism pathway modules, and 1,356 regulatory pathways from KEGG. Our analysis centers on evaluating the performance of GPT-3.5-turbo and Galactica in predicting the outcomes of biochemical reactions based on enzymes and in reconstructing complete pathways or regulatory networks from initial stimuli and enzymes. The results suggest GPT exhibits promising potential in mapping comprehensive pathways documented in KEGG. When introduced to novel scenarios beyond its training scope, GPT is adept at outlining various plausible downstream pathway progressions. This capability not only provides valuable insights but also charts out prospective research trajectories for scientists. However, GPT's current limitation lies in its inability to predict the precise targets involved in the pathway.

To sum up, our contributions are: (1) We curated and assembled 11,944 reaction and 1,836 pathway data from KEGG, enriched with molecular structural and functional insights from UniProt and PubChem, for easier accessibility; (2) We initiated the exploration of the capabilities of large language models for predicting biological pathways. It is important to note that LLMs are designed to be knowledgeable across a vast spectrum of fields. This enables them to synthesize information from diverse areas and offer insights that might be challenging for individual researchers to arrive at. Our findings lay both the strengths and limitations of LLMs, charting a path for further refinements.

## 2 Dataset Construction

Our primary data source was the KEGG database [13]. To assess the efficacy of LLM in predicting biochemical reactions, we compiled a set of 11,944 reactions from KEGG. The constituents of these reactions can be either compounds or glycans. For compounds, their corresponding SMILES were fetched from PubChem using the CAS numbers provided by KEGG. In the case of glycans, we gathered monosaccharide compositions directly from KEGG. It's worth noting that the enzymes responsible for these reactions are occasionally specified. For every unique EC number, we also sourced functional, activity, and sequence information from UniProt when a UniProt ID was available. We consolidated all this data into a structured JSON file.

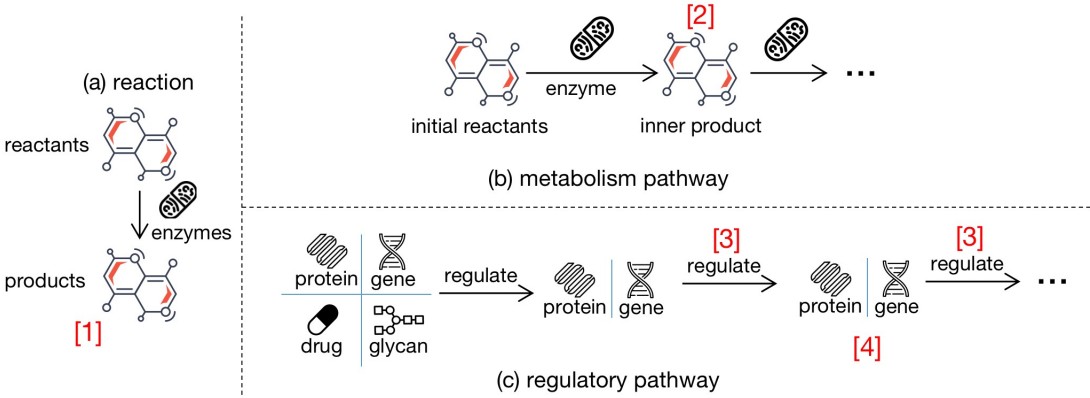

Figure 1: An illustration of biochemical reactions, metabolism pathways, and regulatory pathways. (a) In biochemical reaction prediction, product[1] is predicted; (b) In metabolism pathway prediction, all the inner products[2] are predicted; (c) In the regulatory pathway, the regulated components[4] and the regulatory relationship[3] are predicted.

Additionally, 480 KEGG metabolism pathway modules are collected from the KEGG database through the API. Each of these modules represents a segment of a more intricate pathway, depicted through manually created pathway maps. These maps encapsulate molecular interactions, reactions, and relational networks, predominantly encompassing metabolic pathways and cellular processes. We have structured each module based on consecutive reactions. The reactants, products, and associated enzymes for each of these reactions are also archived in a JSON file. Since each metabolism pathway is composed of several reactions, the detailed information can be retrieved from the previously constructed reaction data. For each reaction within a pathway, we identified its parent reactions as well as any external reactants that are not products of any preceding reactions.

Moreover, we utilized 1,356 KEGG networks, which we recognize as regulatory pathways that encompass protein-protein interactions. These regulatory pathways prioritize variations in molecular interaction and reaction networks, manifesting as network variation maps. Examples include gene expression profiles, protein-protein interaction networks, and perturbations in molecular networks related to diseases and drugs. To articulate a regulatory pathway in textual form, we adhered to KEGG's notational conventions, outlining the regulatory relationships between two components. We follow the KEGG instructions and translate the symbols used in describing the molecular interactions as: **-|** inhibition; **=>** expression; **==** complex formation; **//** missing interaction or reaction; **->** Activation or Enzymatic reaction or transport process; **–** Substrate binding to enzyme or transporter; **»** Enzyme-enzyme relation of successive reactions; **=|** repression.

The detailed information we retrieved from the database is illustrated in Figure 3.

# 3 Prompt Design and Preliminary Results

We used the GPT-3.5-turbo (175 B parameters) and Galactica base model (1.3 B parameters) as our primary models. The detailed prompts are given in Table 2 in Appendix C.

## 3.1 Reaction

Since single reactions are the basic units of a complex pathway, we first assessed a total of 11,756 individual biochemical reactions. However, we excluded those KEGG reactions for which the API retrieval did not provide a list of reactants. If the KEGG entry includes the "ENZYME" section, we incorporate the enzyme information. For predictions made using Galactica, we employed the Question Answering mode: **Question:** *prompt* **\n\nAnswer:**, with the base prompt mirroring that of GPT.

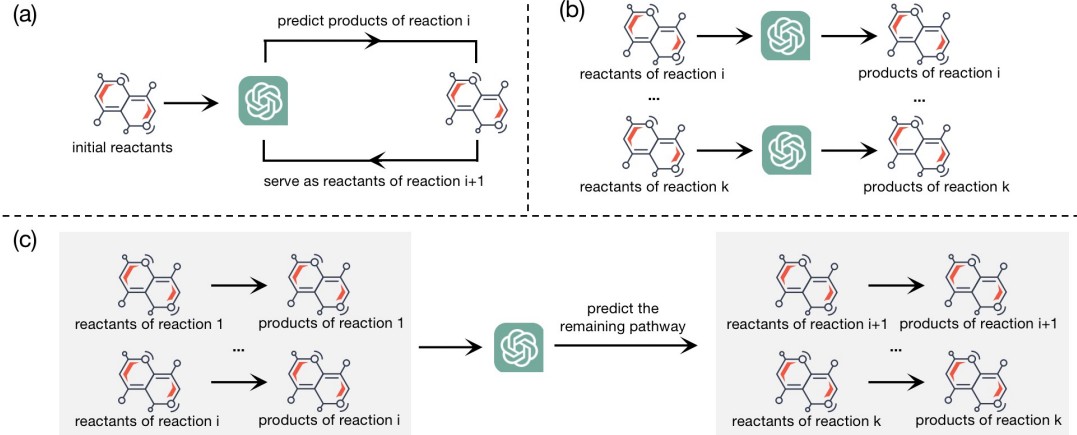

Figure 2: Visualization of three metabolism pathway prediction methods: (a) Auto-regressive prediction, where the LLM predicts subsequent steps based on prior predictions; (b) Auto-regressive prediction with ground truth reactants, where the LLM is provided the true reactants at each step; and (c) One-time prediction, where the LLM is supplied with complete reactions from the first half of the pathway to predict the remaining pathway.

**Evaluation** Predictions were manually assessed by aligning them with the actual products. A prediction is correct if it matches either the name or the SMILES representation of every product, recognizing that certain products might lack a SMILES notation, such as proteins. The results are shown in Table1 1

## 3.2 Metabolism Pathway

We assessed a total of 131 KEGG pathway modules among the 480 collected modules. For each module, we designated the reactants of the initial reaction as the stimuli that initiate the pathway. We applied three different ways to run the experiments, which are also visualized in Figure 2: (1) Using an autoregressive approach, we allowed the large language model to predict subsequent reactions. This meant that its earlier predicted parents reactions were continuously fed back into the model, serving as the prior state of the pathway. The model also needs to decide which products participate in the following reaction. The results are labeled as metabolism_pathway_auto; (2) Similar to the first method, we also conducted experiments where the models were provided with ground truth reactants at each step, rather than relying on their prior predictions, which are labeled as metabolism_pathway_auto_true; (3) For each module, we provide the complete initial half of the ground truth reactions directly to the model and task it with predicting the remaining reactions. The outcomes of this approach are presented in the metabolism_pathway_one-time column in Table 1.

Given that Galactica lacks conversational memory, for the first two methods, we compiled its predictions for all reaction precursors of a reaction and then supplied these to Galactica, prompting it to forecast the following step using a similar Question Answering framework.

**Evaluation** Considering that the pathway is complex to evaluate, we adopted an additional GPT-3.5-turbo as the evaluator to assess the congruence between the predicted pathway/network and the ground truth. In the third method, we have observed that by supplying the reactions, GPT occasionally directly recognizes the entire pathway without the need for explicit prediction. This phenomenon could explain the significantly higher accuracy observed in the metabolism_pathway_one-time results.

**Baseline** We employed DAVID bioinformatics resources [12] as a baseline method, which, given enzymes, identifies the pathways in which these enzymes participate, ranked by p-values, by searching existing databases like KEGG. Since this tool only supports website interaction and the results need to be manually checked, we tested it with 20 selected metabolism pathways, on which GPT-3.5-turbo has a 50% predicting accuracy. We input all the enzymes listed by KEGG and verified whether

Table 1: Accuracy(%) of GPT-3.5-turbo and Galactica (base)

| Model | Reaction | Metabolism Pathway | | | Regulatory Pathway | | |
|---|---|---|---|---|---|---|---|
| David BR | – | 95% | | | 100% | | |
| | | auto | auto_true | one-time | auto | auto_true | one-time |
| GPT-3.5-turbo | 24.06% | 24.62% | 25.38% | 42.31% | 20.11% | 39.94% | 26.85% |
| Galactica (base) | 1.08% | 0% | 0% | 0% | 0% | 0% | 0% |

the corresponding KEGG pathway was returned by the tool. The results are also presented in Table 1.Given that we tested KEGG data and the system directly fetches data from KEGG, its performance is notably superior.

### 3.3 Regulatory Pathway

We examined 353 regulatory pathways taken from the KEGG network section, encompassing protein interactions and signaling pathways. As with previous evaluations, the initial stimulus, whether a protein or another compound, was provided to the model. To represent it in textual form, we followed KEGG's notational standards, detailing the regulatory relationships between two entities, which is included as background information as shown in the prompts in Table 2 in Appendix C. Correspondingly, three types of experiments are performed, which are similar to the examples given in Figure 2, while it no longer predicts the products but the regulated objects and regulatory relationships. (1) For autoregressive generation, we provided the initial step, noting that a single stimulus could be part of various regulatory pathways. The model then anticipates the subsequent interaction based on its prior prediction. Similarly, for Galactica prediction, we collected all the predictions of precursor steps as the knowledge and let it predict the next step; The results are under column regulatory_pathway_auto in in Table 1; (2) In parallel, we also executed experiments where models were given the true previous step to predict subsequent regulations, which are labeled as regulatory_pathway_auto_true in Table 1; (3) For each pathway, we provide the initial half of the ground truth steps directly to the models and task them with predicting the remaining steps. The results are presented in the regulatory_pathway_one-time column.

**Evaluation** Similarly, we adopted an additional GPT-3.5-turbo as the evaluator to assess the congruence between the predicted pathway/network and the ground truth.

**Baseline** Similar to the metabolism pathway, we tested David Bioinformatics Resources with 20 selected regulatory pathways, on which GPT-3.5-turbo has a 50% predicting accuracy. We input all the proteins involved in each pathway and verified whether the corresponding KEGG pathway was returned by the tool. The results are presented in Table 1.

### 3.4 Case Studies

To evaluate GPT's potential in novel biological pathway prediction, we tested it on two papers that were published in 2023, after GPT-3.5-turbo was released.

**Case 1** Embryos from various metazoan lineages can induce a state of transcriptional dormancy, allowing development to pause in response to adverse environmental conditions. Collignon et al. [10] elaborate on the mechanics of this process, the Mettl3 mediating N6-methyladenosine RNA methylation directly destabilizes the mRNA of N-Myc, the transcriptional amplifier, causing suppression of global nascent transcription. This pathway together with the Mettl3 regulating the destabilization of global mRNA leads to a developmental pause.

To assess the capabilities of GPT-3.5-turbo in understanding and predicting this biological process, we presented the following prompt:

**Under adverse environmental conditions, embryonic stem cells from metazoan lineages enter reversible states of developmental pausing, or diapause. There is a pathway starting with Mettl3 regulating the process, predict the pathway starting with Mettl3.**

The response generated by GPT can be referenced in Appendix D.1. Notably, the third point highlighted by GPT accurately indicates that Mettl3 facilitates the addition of m6A modifications to mRNA molecules. However, it doesn't explicitly mention the modification of N-Myc mRNA.

**Case 2** Another study [2] shows that E4ligase, yeast Ufd2 and human UBE4B (the human homolog of Ufd2) move to mitochondria and ubiquitylate mitofusins, thereby inhibiting mitochondrial fusion under environmental stress conditions. This progress leads to mitochondrial fragmentation. Based on this fact, we formed the following prompt for GPT:

**In human cells, under the stress of heat shock, UBE4B translocates to mitochondria, predict the following reaction in this pathway.**

The response generated by GPT can be referenced in Appendix D.2. The second point mentions that UBE4B will interact with Mitochondrial Proteins related to mitochondria maintenance, repair, and quality control, while it doesn't give specific targets. In a real situation, scientists might be then interested in knowing which proteins in mitochondria can interact with UBE4B. We further asked GPT about the details:

**You mentioned that UBE4B can interact with other mitochondria proteins involved in mitochondrial quality control, maintenance, and repair. Can you give some possible proteins in detail?**

The complete answer is given in Appendix D.2. The sixth point mentions that UBE4B could potentially influence the Mfn1 and Mfn2 ubiquitination, resulting in an impact on the mitochondria fusion. This is what the paper suggests and thus it can be considered as a successful prediction.

## 4 Conclusion

In our study, we combined traditional biology with advanced artificial intelligence, specifically focusing on models like GPT-3.5-turbo and Galactica, to explore their potential in biological pathway prediction and analysis. Our results show that GPT can effectively understand and predict complex biochemical interactions, often with a level of accuracy comparable to well-established databases like KEGG. However, they sometimes struggled with specifics, indicating areas that need further improvement. However, Galactica is less effective in handling the long biology description and given comprehensive pathway predictions.

Our real-world tests, using recent scientific publications, highlighted both the strengths and limitations of these models in predicting novel pathways. Leveraging GPT's capacity to amalgamate vast data from diverse sources, it stands as a specialist across multiple biology disciplines, offering holistic predictions on pathways. This serves as a brainstorming tool for scientists, assisting in refining their research focus. However, given its occasionally broad outputs, human intervention remains pivotal to discern and extract the most insightful information from its analyses.

In summary, our findings support the idea that LLMs can be a powerful tool to complement traditional biology research. However, human expertise remains essential to ensure the accuracy and relevance of LLMs predictions. As we move forward, the collaboration between human experts and AI will be crucial for gaining deeper insights into biological systems.

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

## A  Related Works

**Chemical and biochemical reactions prediction** The field of forward chemical reaction prediction predominantly focuses on predicting the resultant products from given reactants. The majority of existing methods amalgamate established templates or patterns with various innovative techniques such as machine learning [7], graph-based idealized molecular orbitals [15], and neural networks [29]. Another significant avenue in this domain is retrosynthesis prediction, which is dedicated to inferring chemical reactants from known products. According to a comprehensive outline by Vignesh Ram Somnath et al.[23], machine learning and deep learning models applicable in this context can be broadly classified into Template-based[9], Template-free [30], and Semi-Template-based models [23].

However, the complexity escalates when it comes to biochemical reactions, which frequently involve large molecules and a myriad of enzymes. A noteworthy attempt to address this was made by BNICE [11], employing a method wherein enzymatic catalysis rules were manually constructed based on their Enzyme Commission numbers. Nonetheless, the approach faces inherent limitations due to the extensive variety and abundance of enzymes in biological systems.

**Biological pathway analysis** In the realm of bioinformatics, a prevalent approach for identifying pathways involves comparing the given proteins or enzymes, which can be deduced from the genome, against similar pathways in existing databases. Notable examples of this approach include PathoLogic [14], PathPred [19], and DAVID bioinformatics resources [12]. Recently, there are also some work in leveraging deep learning for predicting the types of metabolic pathways molecules might follow, based on their structural attributes [3] [4]. Nevertheless, the exploration and application of deep learning for comprehensive pathway prediction remain relatively untapped fields, presenting ample opportunities for discovery and advancement.

**LLMs in pathway prediction** In a previous study, Gilchan Park et al.[21] assessed the capabilities of several prominent large language models, such as Galactica, LLaMA, Alpaca, RST, BioGPT-Large, and BioMedLM, for recognizing protein interactions, pathways, and gene regulatory relations. Their dataset, sourced from the STRING, KEGG, and INDRA databases, facilitated tasks where these LLMs were challenged to predict biological knowledge about protein interactions, identify genes participating in specific pathways, and discern regulatory relationships between genes. Their findings indicate that the latest state-of-the-art LLMs show potential in elucidating biological knowledge. Nonetheless, this study primarily examines the relationship predictions between pairs of proteins or genes. This approach overlooks the frequently encountered and intricate scenarios where a comprehensive pathway needs to be discerned based on pre-existing knowledge of stimuli and reactants.

## B  Dataset structure

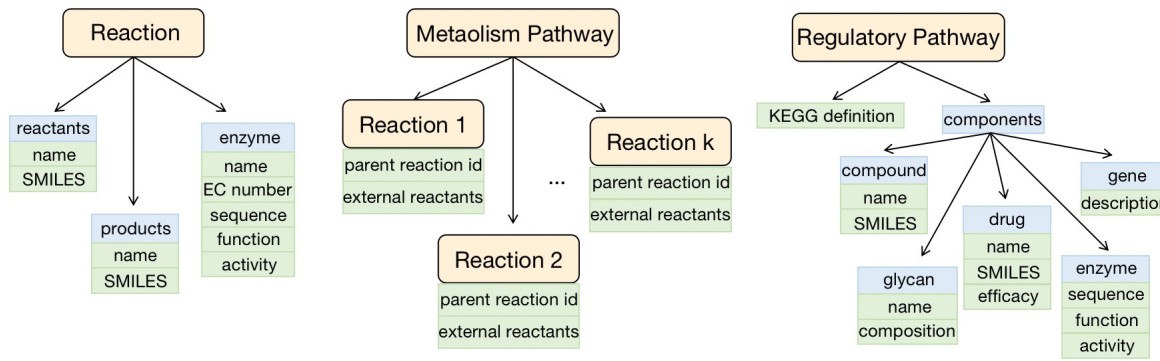

Figure 3: Organized Data Dtructure

## C Prompts

The prompts are shown in Table 2

Table 2: Prompts

| Experiment | Setup Prompts | Initial Steps | Following Steps |
|---|---|---|---|
| Reaction | You are an expert in biology and bio-chemistry. | What are the products of the reaction [REAC-TANT 1] (SMILES) + [REACTANT 2] (SMILES) + ..., under the catalysis of the following enzymes: [ENZYME 1], whose function is described as ...? | – |
| Metabolism Pathway | You are an expert in biology and bio-chemistry. Your job is to predict all the reactions in a biological pathway when cells are exposed to [INITIAL REACTANTS], which has in total k reactions. | Predict the starting reaction i in this pathway, which takes the following reactants: [REAC-TANTS]. | After reaction i, predict the following reaction i+1 in this pathway catalyzed by the following enzymes: [ENZYME 1], whose function is described as ... |
| Regularoty Pathway | Your job is to predict all the steps in a molecular interaction/reaction network when cells are exposed to [COMPOUNDS], which has in total k steps. Possible interactions are A inhibits B; A cause the expression of B; A has an unknown relationship with B; Aactivates or, through an enzymatic reaction, results in B; A binds to B; A has an relation of successive reactions with B; A represses B; A forms a complex with B. Cases when components within a bracket are separated by comma indicate any of them can be involved in this step. Cases when components within a bracket are connected by + indicate they work together in this step. | In the initial step i, ..., Some structural or functional information about the involved components are known: the SMILES of [COMPOUND] is ...; the efficacy of [DRUG] is...; the composition of [GLYCAN] is...; the function of [ENZYME] is described as ... | Given step i, predict step i+1. |

## D Case Studies

This section provides complete answers from GPT in case studies 3.4.

### D.1

**Question** Under adverse environmental conditions, embryonic stem cells from metazoan lineages enter reversible states of developmental pausing, or diapause. There is a pathway starting with Mettl3 regulating the process, predict the pathway starting with Mettl3.

**Answer** Mettl3 (Methyltransferase-like 3) is an RNA methyltransferase that plays a role in RNA modification, specifically N6-methyladenosine (m6A) methylation. It has been implicated in various

cellular processes, including stem cell differentiation. In the context of embryonic stem cells entering reversible states of developmental pausing or diapause under adverse environmental conditions, here's a prediction of the pathway starting with Mettl3:

1. Environmental Stress: Adverse environmental conditions, such as nutrient scarcity, oxidative stress, or other external challenges, trigger a stress response in embryonic stem cells.

2. Mettl3 Activation: In response to the environmental stress, Mettl3 activity may be modulated. Mettl3 is known to be involved in the regulation of RNA stability and translation efficiency through m6A methylation.

3. m6A Modification: Mettl3 catalyzes the addition of m6A modifications to specific RNA molecules, including mRNAs and non-coding RNAs.

4. Altered Transcriptome: m6A modifications can influence the stability, translation, and splicing of target RNAs. In the context of environmental stress, m6A modifications mediated by Mettl3 could lead to changes in the transcriptome of embryonic stem cells.

5. Regulation of RNA Stability: m6A modifications can enhance the stability of specific transcripts, allowing the cell to preserve essential genes for future use during or after the stress period.

6. Translation Regulation: Mettl3-mediated m6A modifications can also affect the translation efficiency of target mRNAs. Some transcripts may be preferentially translated to produce proteins that are needed for stress adaptation.

7. Induction of Developmental Pausing or Diapause: The changes in the transcriptome influenced by Mettl3-mediated m6A modifications may include the regulation of genes associated with the developmental pausing or diapause state. This can lead to the induction of reversible states where the stem cells temporarily halt their differentiation and growth.

8. Cellular Adaptation: During the period of developmental pausing or diapause, embryonic stem cells adapt to the adverse conditions. They conserve energy, maintain stemness, and reduce metabolic demands.

9. Monitoring of Stress Resolution: While in a paused state, stem cells continually monitor the environmental conditions for signs of improvement. If conditions become favorable again, the cells can exit the pausing state and resume normal development.

10. Exit from Diapause: When the adverse environmental conditions subside, Mettl3 and other regulatory factors may reverse the m6A modifications on key transcripts, allowing the stem cells to exit the diapause state.

11. Resumption of Differentiation and Growth: Upon exiting diapause, embryonic stem cells can reinitiate their differentiation programs and resume growth and development.

It's important to note that this is a generalized prediction based on the known functions of Mettl3 and the concept of reversible developmental pausing or diapause in stem cells. The specific targets of Mettl3 and the regulatory mechanisms involved can vary depending on the type of environmental stress and the specific cellular context. To gain more detailed insights, it's advisable to refer to specific research studies or literature related to Mettl3's role in stem cell biology and stress responses.

**D.2**

**Question** In human cells, under the stress of heat shock, UBE4B translocates to mitochondria, predict the following reaction in this pathway.

**Answer** When UBE4B (Ubiquitin-Conjugating Enzyme E2 Variant 4B) translocates to mitochondria in response to heat shock stress in human cells, it suggests a specific response to mitochondrial dysfunction or damage. Predicting the "i+1" reaction in this pathway involves considering the immediate downstream consequences of UBE4B's localization to mitochondria:

1. Translocation of UBE4B: Under heat shock stress, UBE4B translocates from its typical cellular location to the mitochondria.

2. Interaction with Mitochondrial Proteins: Upon reaching the mitochondria, UBE4B may interact with specific mitochondrial proteins, particularly those involved in mitochondrial quality control, maintenance, and repair.

3. Ubiquitination of Mitochondrial Proteins: UBE4B, being an E3 ubiquitin ligase or working in conjunction with other E3 ligases, may ubiquitinate target proteins within the mitochondria. This ubiquitination could serve as a signal for the degradation or modification of these mitochondrial proteins.

4. Activation of Mitophagy: The ubiquitination of mitochondrial proteins can trigger mitophagy, a selective form of autophagy aimed at removing damaged or dysfunctional mitochondria. This is an essential process for maintaining mitochondrial health.

5. Mitochondrial Proteostasis: UBE4B's translocation and subsequent ubiquitination of mitochondrial proteins may also be involved in maintaining mitochondrial proteostasis, ensuring that misfolded or damaged proteins are properly processed or eliminated.

6. Stress Response Activation: UBE4B's translocation to mitochondria and its involvement in mitochondrial quality control may activate cellular stress responses, such as the unfolded protein response (UPR) or the heat shock response. These responses can help the cell cope with mitochondrial stress and prevent further damage.

7. Mitochondrial Function Modulation: Depending on the specific mitochondrial proteins targeted by UBE4B, mitochondrial function may be modulated. This can include changes in energy production, ROS generation, and the overall health of the mitochondria.

8. Cellular Consequences: The downstream consequences of UBE4B translocation to mitochondria can have a significant impact on cellular homeostasis. Mitochondrial health is crucial for energy production and various cellular processes, so disturbances in this system can affect cell viability and function.

9. Adaptive or Cell Death Pathways: Depending on the extent of mitochondrial damage and the success of stress responses, the cell may adapt to the stress or, in severe cases, undergo programmed cell death (apoptosis) to prevent further harm to the organism.

It's important to note that the specific mitochondrial proteins targeted by UBE4B and the precise downstream effects can vary depending on the cell type, the severity of the heat shock, and the context of the stress response. This prediction provides a general overview of the pathway's progression following UBE4B translocation to mitochondria in response to heat shock stress. For detailed insights, it's advisable to refer to specific research studies or literature related to UBE4B's role in mitochondrial quality control and cellular stress responses.

**Question** You mentioned that UBE4B can interact with other mitochondria proteins involved in mitochondrial quality control, maintenance, and repair. Can you give some possible proteins in detail?

**Answer** UBE4B is an E3 ubiquitin ligase known to interact with various proteins involved in different cellular processes, including mitochondrial quality control, maintenance, and repair. While I can provide some examples of potential proteins that UBE4B might interact with within the context of mitochondria, please note that these interactions can be complex and context-dependent. Here are some possible proteins:

1. Parkin (PARK2): Parkin is an E3 ubiquitin ligase that plays a crucial role in mitophagy, the process by which damaged or dysfunctional mitochondria are targeted for degradation. UBE4B could potentially interact with Parkin to regulate mitophagy or the ubiquitination of mitochondrial proteins.

2. PINK1 (PTEN-induced kinase 1): PINK1 is a mitochondrial kinase involved in monitoring mitochondrial health. It accumulates on the surface of damaged mitochondria and recruits

Parkin to promote mitophagy. UBE4B might interact with PINK1 or modulate its activity in response to heat shock stress.

3. Hsp70 and Hsp90 Chaperones: Heat shock proteins, such as Hsp70 and Hsp90, are involved in protein folding and refolding, particularly during stress conditions like heat shock. UBE4B could interact with these chaperones to facilitate the refolding or degradation of misfolded mitochondrial proteins.

4. Mitochondrial Import Machinery: Proteins involved in the import of mitochondrial proteins might interact with UBE4B during the translocation process. For example, TOM (Translocase of the Outer Mitochondrial Membrane) and TIM (Translocase of the Inner Mitochondrial Membrane) complexes facilitate the import of proteins into mitochondria. UBE4B may interact with these complexes or their associated chaperones to regulate mitochondrial protein import.

5. Mitochondrial Proteases: Proteases within the mitochondria, such as Lon protease and ClpXP protease, are responsible for degrading damaged or misfolded proteins. UBE4B may interact with these proteases to target specific substrates for degradation.

6. Mfn1 and Mfn2 (Mitofusins): These proteins are involved in mitochondrial fusion, a process that can help mitigate mitochondrial damage by mixing the contents of healthy and damaged mitochondria. UBE4B could potentially influence the ubiquitination status of Mfn1 and Mfn2, impacting mitochondrial fusion.

It's important to emphasize that these interactions are hypothetical and based on the known functions of UBE4B and the proteins listed above. The specific interactions and roles of UBE4B in mitochondrial quality control during heat shock stress would require experimental investigation to determine their precise mechanisms and significance in the context of cellular stress responses.

