# OpenReview forum: "ChatPathway: Conversational Large Language Models for Biology Pathway Detection"
_NeurIPS.cc/2023/Workshop/AI4Science — NeurIPS2023-AI4Science Poster_

### Official Review · Reviewer_n34m · 2023-10-24
**Interesting case studies, but lack comprehensive analysis**

**Rating:** 5
**Confidence:** 3

**Review:**

The authors present ChatPathway, a proof-of-concept study on using LLMs for predicting biochemical reactions and pathways. The main contribution of this work, I think, is that the authors have curated and assembled 11,944 reaction and 1,836 pathway data from KEGG, enriched with molecular structural and functional insights from UniProt and 70 PubChem, for easier accessibility. They use such data evaluate LLMs for three tasks: (1) biochemical reaction prediction; (2) metabolism pathway prediction; and (3) regulatory pathway prediction. The authors found that GPT-3.5 demonstrated strengths in pathway mapping, but Galactica encountered challenges

Specific comments:
1. Not enough comparisons. The authors only evaluated GPT-3.5 and Galactica, without other LLMs. The authors might also want to experiment with GPT-4, LLaMA, as well as biomedical LLMs such as BioGPT. More comparisons can draw more comprehensive results.
2. Will the curated dataset be publicly available? I didn't see any supplementary materials or GitHub links.
3. The DAVID tool shows much better performance than LLMs, so why not augment LLMs with such a tool?
4. It would be nice to also provide some error analysis (e.g., on their types and percentages), since both LLMs do not performance well.

---

### Meta-Review · Area_Chair_9fj1 · 2023-10-26

**Recommendation:** Accept (Poster)
**Confidence:** 4

**Metareview:**

Good paper.
Accept.